# Strengthening and Toughening CNTs/Mg Composites by OpTimizing the Grinding Time of Magnesium Powder

**DOI:** 10.3390/nano12234277

**Published:** 2022-12-01

**Authors:** Yunpeng Ding, Yizhuang Zhang, Zhiyuan Li, Changhong Liu, Hanying Wang, Xin Zhao, Xinfang Zhang, Jilei Xu, Xiaoqin Guo

**Affiliations:** School of Materials, Zhengzhou University of Aeronautics, Zhengzhou 450046, China

**Keywords:** carbon nanotube, magnesium matrix nanocomposite, strengthening, toughening, mechanical property

## Abstract

In this paper, CNT/Mg composites with high compressive properties were prepared by using Ni-plated CNT and pure magnesium powder as raw materials through the grinding of magnesium powder, ball-milling mixing and hot-pressing sintering. The effect of grinding time for finer magnesium powder on the microstructure and properties of the final composites was studied mainly by SEM, XRD, HRTEM and compression tests. The results show that with the prolongation of milling time, the magnesium particle size decreases gradually and the CNT dispersion becomes more uniform. Moreover, the nickel layer on the surface of CNT reacts with highly active broken magnesium powder in the sintering process to generate MgNi_2_ intermediate alloy, which significantly improves interface bonding. The strength and fracture strain of composites are significantly increased by the combined action of the uniform distribution of CNTs and strong interface bonding from the MgNi_2_ phase. The compressive strength, yield strength and fracture strain of the composites, prepared with a 60 h grinding of magnesium powder, reached 268%, 272% and 279% of those in composites without the grinding of magnesium powder.

## 1. Introduction

Magnesium matrix composites have the advantages [1] of low density, high specific strength and high elastic modulus, overcoming the bottleneck of low absolute strength and low stiffness in magnesium alloys, and thus, magnesium matrix composites have great application potential in transportation, aerospace and 3C industries under the trend of carbon neutrality [2].

As an important member of the fullerene family, carbon nanotube (CNT) is synthesized by methods, such as chemical vapor deposition and microwave irradiation [3]. CNT has super high mechanical properties [4], high heat conductivity [5] and low density, making it the most potential reinforcement for magnesium matrix composites [6]. However, the large Van der Waals force of CNTs makes them easy to agglomerate and difficult to disperse evenly in the matrix [7]. Moreover, due to the poor wettability between CNT and metal, the interface between CNT and magnesium matrix is poor. The combined effects of these two factors lead to the unsatisfactory strength improvement of CNTs reinforced magnesium matrix (CNTs/Mg) composites, and the toughness is difficult to improve cooperatively [8], which hinders its development and application. 

Recently, many studies on improving the dispersion of CNT mainly focus on the liquid dispersion method (ultrasonic treatment, mechanical stirring, adding surfactant and so on), ball milling [9], surface plating, stir casting [10], friction stir welding [11], in situ synthesis of CNT, etc. These methods have achieved good results. Among these methods, the ball-milling method with inert gas protection is easy to operate and has low equipment requirements. Studies on improving the bonding strength of CNT/Mg interface mainly consist of surface plating of CNT (Ni [12], W, SiO_2_, TiO_2_ [13], MgO [14], SiC [15], Si [16], etc.), CNT doping (boron doping, etc.), CNT mixing with Al and then adding it into Mg as a new phase [17], etc. The method of plating metal layer on CNT surface is simple and effective [18]. With the enthusiasm for bionics [19], Yeyang Xiang et al. [20] constructs CNTs/Mg micro-nano-layered composites by electrophoretically depositing of CNT layer on Mg foils and subsequent rolling to optimize the distribution of CNT. 

Considering these above factors, nickel-plated CNTs were used in this study to improve interfacial bonding, and CNTs were dispersed in matrix by ball milling. A previous study [18] has shown that the grinding and refining of matrix powder before mixing with CNT can improve the dispersion of CNT. However, the grinding process and mechanism are still unclear. Therefore, in this paper, when nickel-plated CNTs and pure magnesium powder were used to prepare CNTs/Mg composites, the magnesium powders were first ground and refined, and then mixed with CNTs and sintered by vacuum hot pressing. The influence law and mechanism of grinding time on microstructure and mechanical properties of composites were studied. The purpose of this study is to solve the problem of poor dispersion and interface bonding of CNT in magnesium matrix composite contributing to the application of extended magnesium matrix composites. 

## 2. Materials and Methods

As-received materials are nickel-plated CNTs (CNT content > 38 wt%, Ni content > 60 wt%, JCMWNI4, Jiacai Technology Co., Ltd., Chengdu, China) with 30–50 nm outer diameter of and <10 μm length and pure magnesium powder (20μm, Yaotian new material Co., Ltd., Shanghai, China). The commercial CNTs were prepared by the CVD method. Then, a layer of Ni particles was plated on the surface of CNTs by electroless plating process. In the preparation process of CNTs/Mg composites, the magnesium powder was first ground, then mixed with CNT and sintered by vacuum hot pressing. 

During the grinding process, magnesium powders were grinding for 0 h, 15 h, 30 h, 45 h and 60 h, respectively, at a speed of 200 r/min in a ball mill using zirconia balls. The ball-to-material ratio was 10:1. Stearic acid was added as a ball milling control agent, and the amount of stearic acid was 1% per 15 h of milling. Then, the refined magnesium powder obtained was mixed with 1.0 vol.% CNTs by ball milling at a speed of 300 r/min for 2 h. The powder mixture is then placed into a graphite mold (30 mm inner diameter) in a vacuum hot-pressing sintering furnace (Centorr Vacuum Industries, Nashua, NH, USA). Samples were sintered at 500 °C for 1 h with a pressure of 40 MPa. The heating rate was 10 °C/min. When the temperature was heated to 400 °C, the temperature was kept for 2 h to make the stearic acid fully volatilized. After sintering, it was cooled in the furnace. For comparison, as-received pure Mg sample was also fabricated using the same consolidation conditions. 

The mixed powders were characterized by X-ray diffraction (XRD, Rigaku SmartLab, Tokyo, Japan). The sintered specimens and mixed powders were characterized using a high-resolution transmission electron microscope (HRTEM, TECNAI G2 F20S-TWIN, FEI company, Hillsboro, OR, USA) and scanning electron microscope (SEM, JSM-7001F, JEOL, Tokyo, Japan). TEM images were analyzed and processed by Digital Micrograph software. The microstructure of samples was observed using an optical microscope (OM, BX51, Olympus, Tokyo, Japan). The sintered samples were cut into rectangular-shaped specimens of 12.5 mm × 5 mm × 5 mm and the compression test was undergone on mechanical testing equipment (CMT5305, Suns, Shenzhen, China) under a compression speed of 0.0625 mm/min. Vickers hardness measurement was carried out on a digital Vickers hardness tester (TH700, Beijing Times Peak Co., Ltd., Beijing, China) with a load of 0.1 kgf and a holding time of 15 s. 

## 3. Results

### 3.1. Morphology

Figure 1 shows the SEM images of magnesium powder after grinding at different times. In Figure 1a, the original magnesium particles are spherical with a mean size of 20 μm and the surface of the magnesium powder particles is relatively smooth. As can be seen from Figure b–f, with the prolonging of the grinding time, magnesium powders weld and fracture with each other under the action of impact and friction energy of milling [21]. The size of magnesium particles gradually decreases from 20 μm to about 3 μm. Their shapes changed from spheres to irregular flakes. In this way, the specific surface area of magnesium powder gradually increases, and the CNTs are easier to disperse evenly in the subsequent mixing process.

Figure 2 shows the SEM morphology of nickel-plated CNTs. The original CNTs were treated with nickel plating. After electroless nickel plating, nickel is deposited on the surface of CNTs in the form of particles about 30 nm in size. The nickel layer can further improve the wettability of the CNT and magnesium matrix, which enhance the bonding strength of CNT/Mxg interface, so as to obtain higher mechanical properties.

Figure 3 shows the SEM morphology of CNTs/Mg composite powder prepared by mixing 1 vol.% CNTs and magnesium powder grinded for different times. The macroscopic morphology of the composite powder is similar to that of Mg powders after grinding. As can be seen in the high magnification picture, there are CNT aggregates among the particles in the composite powder with a grinding time of less than 30 h (Figure a–f). This means CNTs are not completely uniformly dispersed. With an increase in grinding time, the tendency of CNT aggregating becomes less and less. When the grinding time is longer than 45 h, the CNTs are evenly dispersed on the surface of magnesium particles, and no obvious agglomeration is observed. This is because the size of the Mg particle decreases gradually after a longer grinding time, and the specific surface area increases substantially, which provides a larger space for the dispersion of CNTs. Figure 4 shows the EDS point result of CNTs/Mg composites fabricated from 60 h-grinded Mg powder. The figure shows that CNTs are evenly distributed on the surface of magnesium powder. The components at this point are mainly Mg, C and Ni, which just correspond to magnesium powder and nickel-plated CNTs. It is also found that the length of CNT decreases to about 500 nm. This is because the CNTs were cut short by impact and friction in the process of ball milling.

### 3.2. Microstructure and Phase

Figure 5 shows the XRD pattern of CNTs/Mg composite after hot-pressing sintering. It can be seen that the CNTs/Mg composite exhibits diffraction peaks corresponding to Mg and MgO. Mg is the main phase of composites. In addition, a small amount of the MgO phase is also found in the sintered samples, which may be caused by the inevitable contact reaction between magnesium and oxygen during powder preparation. Another possible reason for the presence of MgO is the in situ reaction between Mg and residual stearic acid (the control agent of milling). Ni and C are not found, mainly because their content is too low to be detected by XRD [18].

Figure 6 shows the microstructure of CNTs/Mg composites prepared with magnesium powder grinded for different times. The grain morphology of the composite prepared from pure magnesium is close to spherical, and the grain size is about 22 μm (Figure a). After 15 h of grinding, the grain shape is irregular and the grain size varies greatly. With the increase in grinding time, the grain size of the matrix gradually decreases and the shape becomes flat. The average grain size was calculated by drawing a line on the diagram and counting the number of grains. According to statistics, with the increase in milling time, the equivalent average grain size is about 22 μm, 11 μm, 6 μm, 5 μm and 3.8 μm, respectively, as shown in Figure 7. The longer the milling time, the better the grain refinement. In terms of the Hall–Petch relationship, this will increase its strength to some extent. 

In order to observe the interface bonding of CNT and magnesium, HRTEM analysis was performed on the composites. Its HRTEM diagrams are shown in Figure 8. The CNT is in the area surrounded by the white dashed line and the atomic layer spacing is 0.34 nm for CNT (002) planar spacing. However, nickel is not observed on the surface of CNT. Instead, MgNi_2_ intermediate alloys (regions B, C and D in Figure a) are found in CNTs/Mg interface with 0.22 nm of (106) planar spacing and 0.20 nm of (114) planar spacing. These results indicate that the Ni plating on CNT surface and Mg matrix undergo a chemical reaction to generate the MgNi_2_ phase. The intermediate alloy is beneficial to improve the wettability between CNT and magnesium matrix. The interfacial bonding strength is significantly improved. In addition, the CNT structure is complete, indicating that the CNT structure is not seriously damaged in the process of ball milling and sintering.

### 3.3. Mechanical Property and Relative Density

Figure 9 shows the effect of grinding time for magnesium powder on the hardness and relative density of CNTs/Mg composites. The relative density of composites increases with the prolongation of grinding time. Additionally, the relative density is slightly more than 100%, which is possibly due to the inevitable contact and reaction between magnesium powder and air during the preparation of composite materials. The reason may also be the reaction between magnesium powder and stearic acid, which has not been completely removed, to generate MgO during sintering. MgO has a higher density than magnesium. As a result, the final density of a composite is slightly greater than 100%. The degree of oxidation should not be very high considering that most of the experiment was conducted in the glove box and vacuum hot pressing furnace. This is consistent with the experimental results of XRD in Figure 5. With the prolongation of grinding time needing more stearic acid, the relative density increased gradually, indicating that the chemical reaction between stearic acid and magnesium plays a more important role in the production of MgO. 

The Vickers hardness of pure magnesium is 41 HV. With an increase in grinding time, the hardness of composites increases gradually. Compared with pure magnesium, the hardness of CNTs/Mg composites using Mg powder grinded for 0 h, 15 h, 30 h, 45 h and 60 h are increased by 40%, 42%, 42%, 63% and 161%, respectively. The maximum hardness of the composite reached 133 HV after 60 h of grinding. However, with a longer grinding time, the significant increase in hardness is mainly related to the strengthening effect of CNTs and the strengthening effect of fine grains in the grinding process. 

According to the Hall–Petch relationship, the effect of fine grain strengthening should be relatively small. This shows that the main strengthening effect is from CNT strengthening. The longer the grinding time, the stronger the strengthening effect of nickel-plated CNTs is. This is because the size of magnesium powder decreases and the surface energy increases after grinding, and the nickel-plated CNTs are easier to disperse evenly and reduce the occurrence of agglomeration. What is more, nickel-plated CNTs and magnesium have good wettability due to the appearance of MgNi_2_, which can form excellent interface strength, and the load can be well transferred through the interface.

Figure 10 shows the effect of grinding time for magnesium powder on the compressive properties of CNTs/Mg composites. The content of CNTs is 1 vol.%. The ultimate compressive strength (UCS), yield strength (YS) and fracture strain (FS) of composites prepared with ungrinded magnesium powder are 188 MPa, 166 MPa and 3.9%, respectively. The strength and plasticity are relatively low. Compared with composites with ungrinded magnesium powder, the UCS, YS and FS of composites prepared by grinded magnesium powder are significantly increased. The YS and UCS of composites increase gradually with a longer grinding time. When the grinding time is 60 h, the UCS, YS and FS of composites reach values of 517 MPa, 460 MPa and 10.9%, respectively, which are 268%, 272% and 279% of the corresponding properties of the composite without grinding of magnesium powder, respectively. With the prolongation of grinding time, the FS of composites first increases and then decreases, and the highest fracture strain is 12.4% at 45 h of grinding. The FS decreases slightly at 60 h to 10.9%. Considering its ultra-high strength, it can be considered that the composite at this time has the best comprehensive mechanical properties. Compared with similar materials reported in the literature [17,22,23,24], the mechanical properties of this study are much higher, especially in terms of strength. Compared with pure magnesium, the strength of CNTs/Mg composites significantly increases and the fracture strain decreases. When the grinding time is 60 h, the compressive strength and yield strength of the composites are 140% and 380% of the corresponding properties of pure magnesium, respectively.

## 4. Discussion

The magnesium powder and zirconia ball collide and rub against each other at high speed during the process of grinding, and the magnesium powder particles are welded and deformed. As magnesium has a hexagonal crystal structure and relatively poor plasticity, when the deformation exceeds a certain degree, the grinding effect [2] would occur. The size of magnesium powder decreases and the specific surface area increases, which is beneficial to the uniform dispersion of CNTs in the matrix. The magnesium powder particles grinded for 60 h shows a thin flake, formed by the accumulation of many smaller planar sheets. Such submicron structure has high surface activity and is more likely to react with the plated nickel of CNT to combine with CNT, thus obtaining high interfacial bonding strength. 

With the longer grinding time, the compressive strength and yield strength of the composites increase gradually, and the plasticity of the composites is higher than that of the ungrinded composites. The main reason is that the large-size magnesium particles have a small surface area and limited dispersion space for CNT with none or less grinding time. As a result, CNTs easily cluster due to their Van der Waals force. The internal bonding force of agglomerated CNTs is weak and the internal CNTs are basically not in touch with the matrix, which cannot play the role of stress transfer [25] and grain-pinning effect. Therefore they cannot play the role of strengthening the matrix. Inversely, the stress near the CNT cluster is easy to concentrate during deformation, which becomes the crack source and causes premature cracking. As a result, the FS and strength of composites are not high. 

Moreover, the submicron structure of grinded Mg particles has high surface activity, and the nickel layer on the surface of CNT reacts with these Mg particles to form MgNi_2_ phase, forming strong interfacial bonding. It makes the stress more easily transferred to the surrounding matrix through the high aspect ratio of CNT, and makes CNT play full role in strengthening. The longer the grinding time is, the more uniform dispersion of CNT is and the greater effect of stress transferring is, resulting in a higher strength. However, when the grinding time is very long, the fracture strain decreases slightly, which may be caused by the increase in stearic acid residue and the increase in MgO content. 

## 5. Conclusions

The poor dispersion and interface bonding of CNT in CNTs/Mg composites limit the development of the composite. The purpose of this study is to solve the problem of poor dispersion and interface bonding of CNT in magnesium matrix composite contributing to the application of extended magnesium matrix composites. In this study, the grinding of magnesium powder was used to refine the particles and improve the dispersion of CNT. Additionally, Ni-plated CNTs were employed in strengthening the interface bonding. The effects of grinding time on the microstructure, interface and mechanical properties of the composites were analyzed. The main conclusions are as follows:

(1) With the prolongation of grinding time, the hardness, ultimate compressive strength, yield strength and fracture strain of composites all increase significantly. The ultimate compressive strength, yield strength and fracture strain of the composite, with magnesium powder grinded for 60 h, reach 268%, 272% and 279% of the corresponding properties of composites prepared with ungrinded magnesium powder.

(2) The mechanical properties of composites are greatly improved because the shape and size of magnesium powder are changed, and the specific surface area is sharply increased after grinding. It is easier to get uniform dispersion for nickel-plated CNTs. On the other hand, the nickel layer on the surface of CNT reacts with Mg to form an MgNi_2_ phase, forming strong interfacial bonding. 

## Figures and Tables

**Figure 1 nanomaterials-12-04277-f001:**
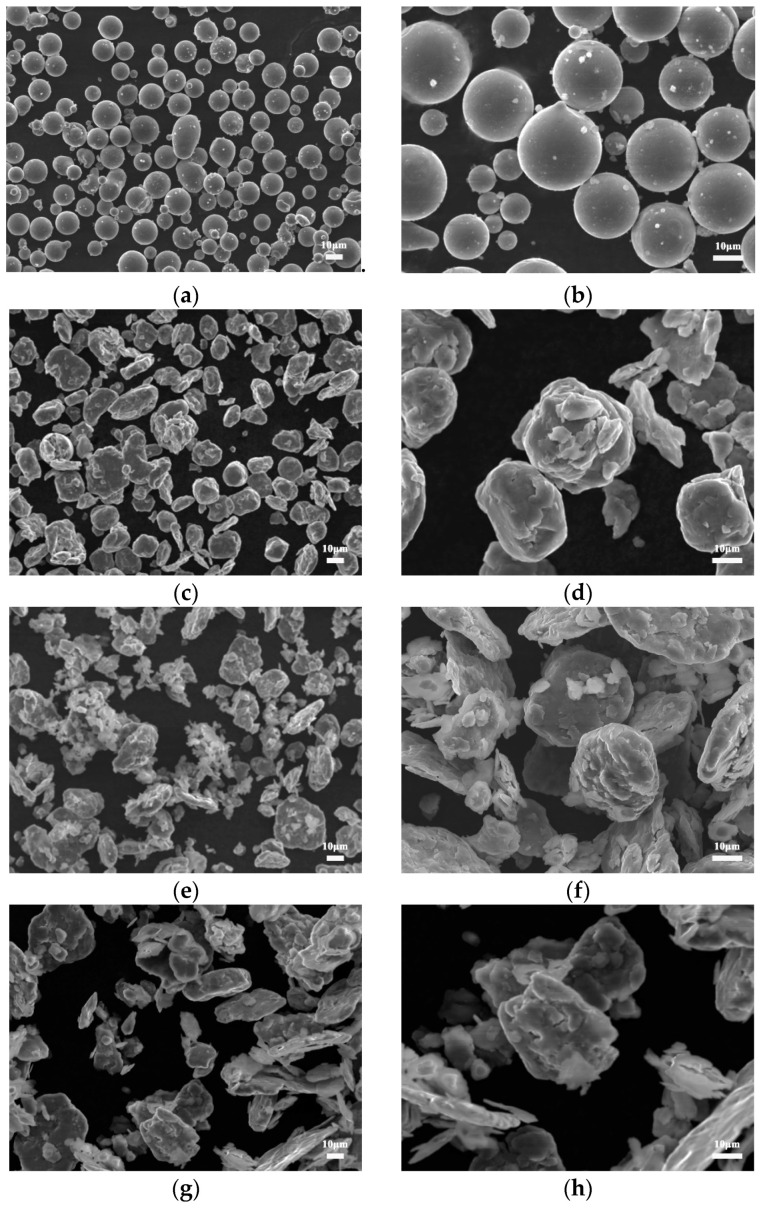
SEM morphology of magnesium powder after grinding for different times. (**a**,**b**) as-received, (**c**,**d**) 15 h, (**e**,**f**) 30 h, (**g**,**h**) 45 h and (**i**,**j**) 60 h.

**Figure 2 nanomaterials-12-04277-f002:**
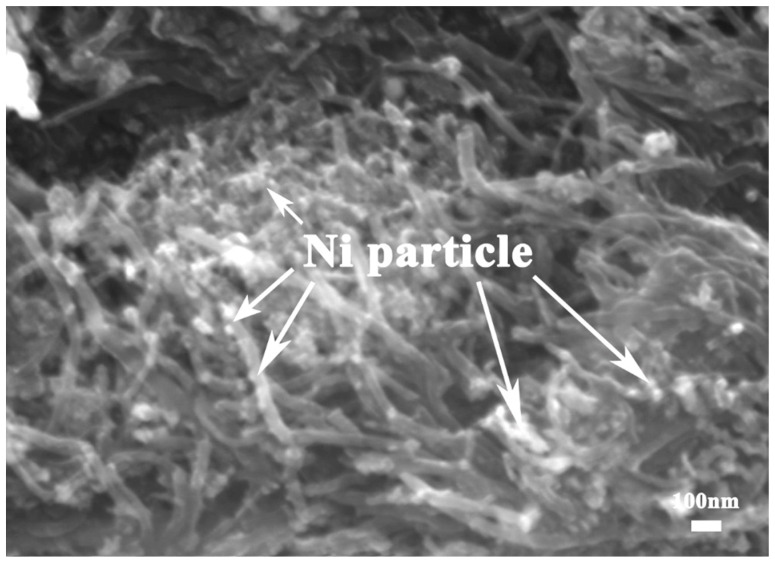
SEM morphology of nickel-plated CNTs.

**Figure 3 nanomaterials-12-04277-f003:**
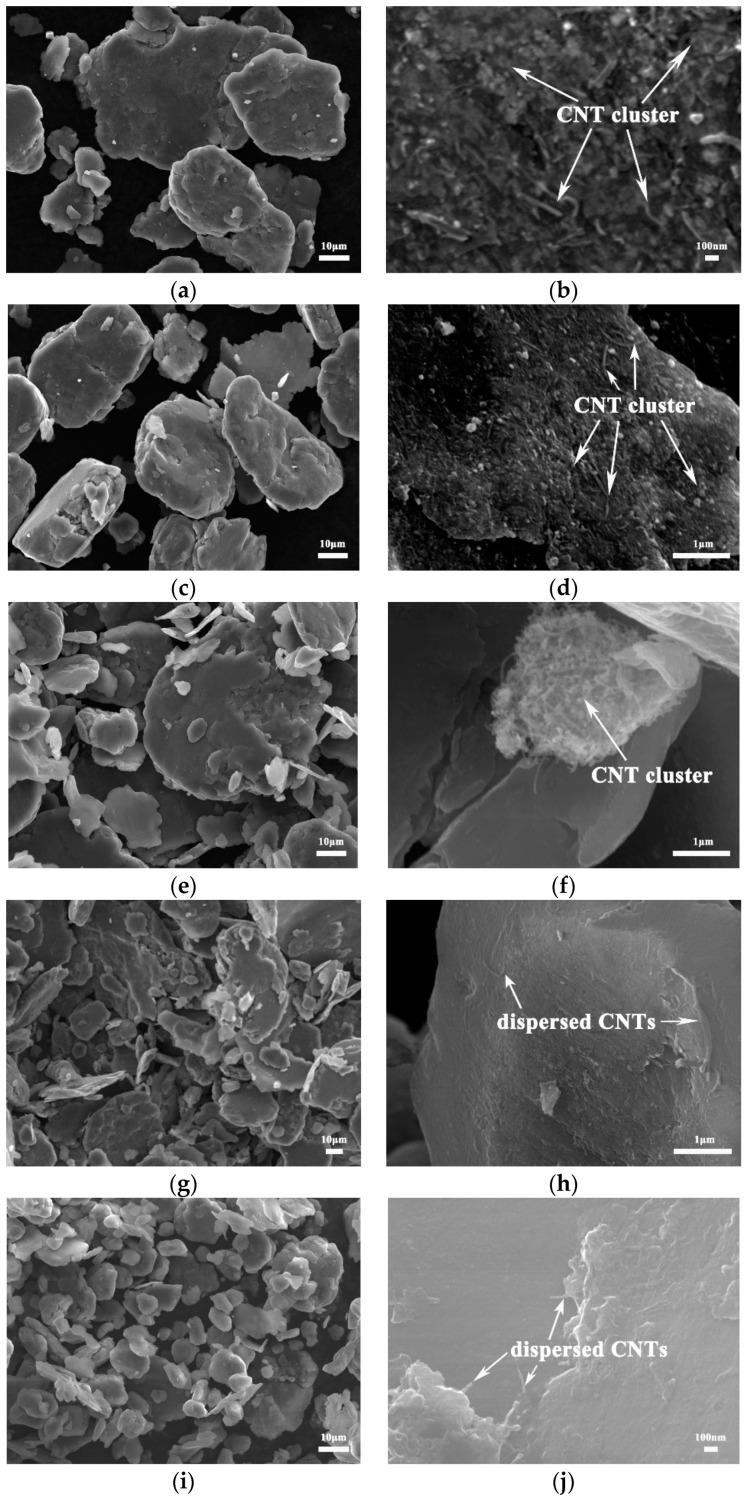
SEM morphology of CNTs/Mg powders with magnesium powders grinded for different times. (**a**,**b**) as-received, (**c**,**d**) 15 h, (**e**,**f**) 30 h, (**g**,**h**) 45 h and (**i**,**j**) 60 h.

**Figure 4 nanomaterials-12-04277-f004:**
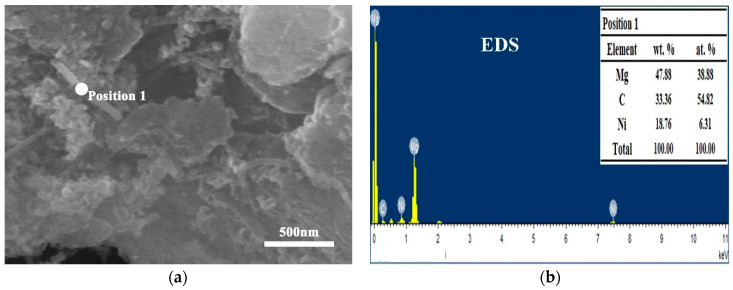
EDS point result from CNTs/Mg composites fabricated from Mg powder grinded for 60 h, (**a**) EDS location and (**b**) EDS result.

**Figure 5 nanomaterials-12-04277-f005:**
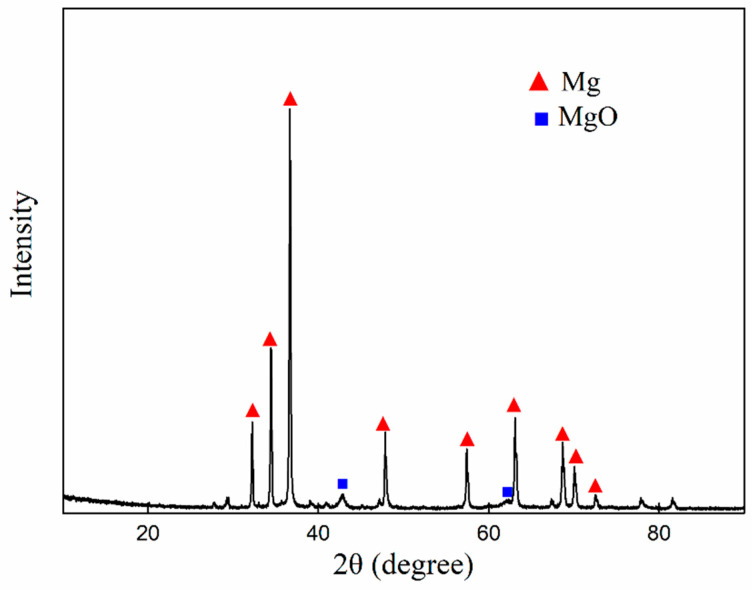
XRD pattern of CNTs/Mg composites after hot pressing sintering.

**Figure 6 nanomaterials-12-04277-f006:**
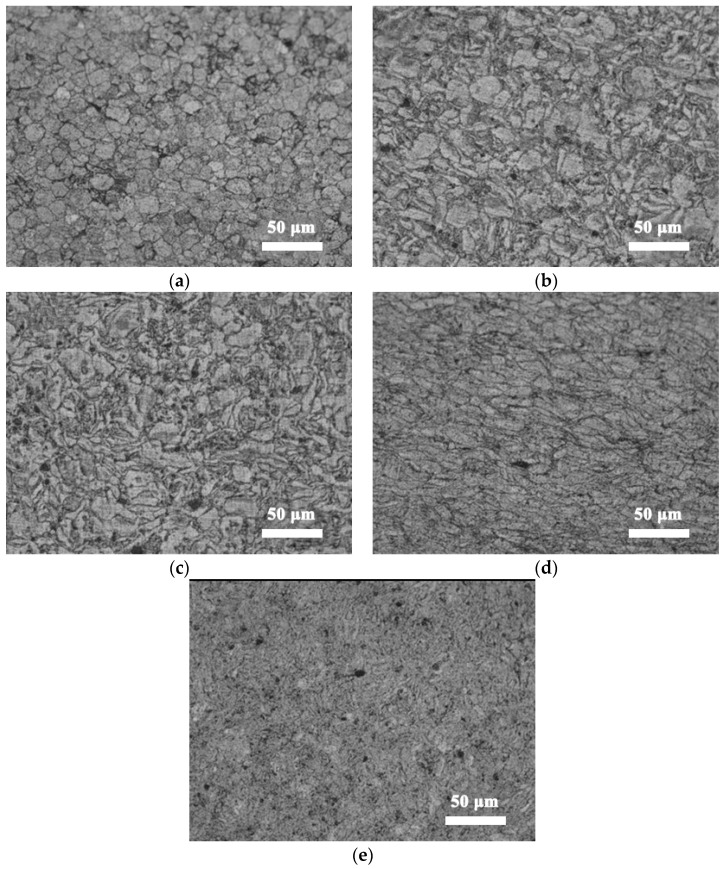
Microstructure of CNTs/Mg composites fabricated from Mg powder grinded for different times. (**a**) 0 h, (**b**) 15 h, (**c**) 30 h, (**d**) 45 h and (**e**) 60 h.

**Figure 7 nanomaterials-12-04277-f007:**
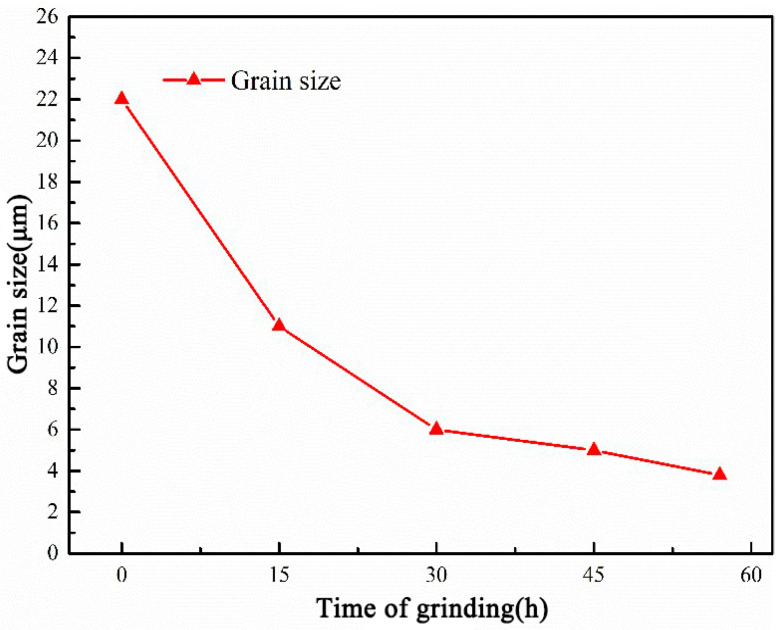
Grain size of CNTs/Mg composites prepared by Mg powder grinded for different times.

**Figure 8 nanomaterials-12-04277-f008:**
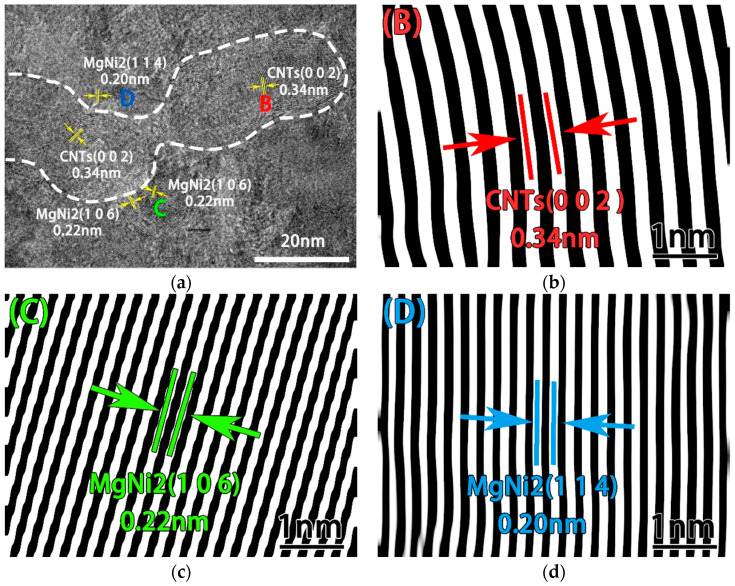
TEM images of CNTs/Mg composites fabricated from Mg powder grinded for 60 h. The layer spacing of regions B, C and D in Figure (**a**) are shown in Figure (**b**–**d**).

**Figure 9 nanomaterials-12-04277-f009:**
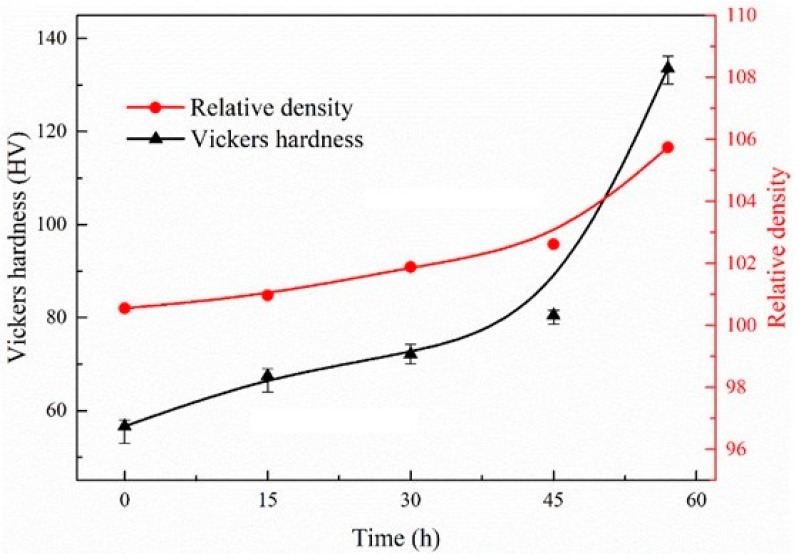
Effect of grinding time for magnesium powder on the hardness and relative density of CNTs/Mg composites.

**Figure 10 nanomaterials-12-04277-f010:**
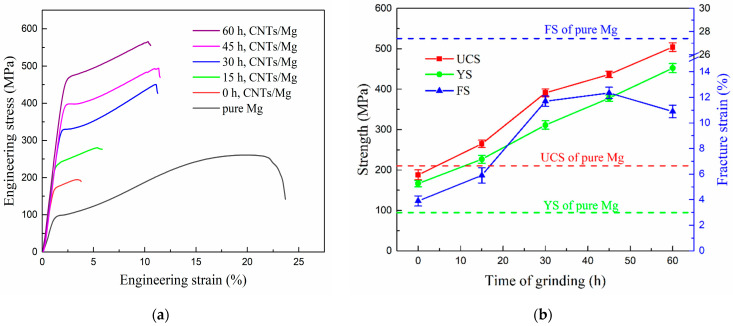
Effect of grinding time for magnesium powder on the compressive properties of CNTs/Mg composites. (**a**) typical engineering stress–strain curve, (**b**) strength and fracture strain.

## Data Availability

The result data are available to download from [https://pan.baidu.com/s/1RnZ2NaJ43R70norsSbmxAg?pwd=kebb] with password of “kebb”.

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
