# Peer review of "Strengthening and Toughening CNTs/Mg Composites by OpTimizing the Grinding Time of Magnesium Powder"

_nanomaterials, 2022, doi:10.3390/nano12234277_

Round 1
Reviewer 1 Report
In this manuscript, the effect of grinding process on the size of Mg powders and the microstructure and mechanical properties of Ni-plated CNT/Mg composite are studied. I suggest the editor to reconsider this manuscript after the author address these issues.
1. The preparation of this manuscript is too slopy. For example, the Fig. 6 and Fig. 8 are exactly the same.
2. The quality of the photo in Figure 2 is poor, although the magnification level of this image is not very high.
3. Although the authors show a large number of SEM images, they do not provide any relevant EDS results. So how did the authors prove that “the dispersion of CNTs is improved”. Please explain.
4. What kind of specimen is used for the TEM characterization? There is no relevant description on this. Please state.
5. Please provide the XRD results of the as-prepared Ni-CNT/Mg composite. Because it can be seen from the TEM results that a large amount of MgNi2 phase is formed.
Reviewer 2 Report
1. Supplement the literature review by methods of synthesis of CNT:
Synthesis of carbon nanotubes using microwave radiation: technology, properties and structure Shchegolkov A.V., Shchegolkov A.V. General Chemical Journal Reference disabled, 2022, 92(6), p.
2. To complete the literature review is an example of the practice and use of different types of MWNTs:
Ali I., AlGarni T.S., Shchegolkov A. et al. Heat-regulating planar electric heaters based on polymers modified with MWNTs. Polym. Bull. 78, 6689-6703 (2021). https://doi.org/10.1007/s00289-020-03483-y
To complete a review of the literature on the method of mechanical dispersion of MWNTs:
Eom, J.Y.; Kim, D.Y.; Kwon, H.S. Effects of ball-milling on lithium insertion into multi-walled carbon nanotubes synthesized by thermal chemical vapor deposition. J. Power Sources 2006, 157, 507-514.
Shchegolkov, A.V.; Jang, S.-H.; Shchegolkov, A.V.; Rodionov, Y.V.; Glivenkova, O.A. Multistage mechanical activation of multi-walled carbon nanotubes in creating electric heaters with self-regulating temperature. Proceedings 2021, 14, 4654. https://doi.org/10.3390/ma14164654.
3. Need to describe in detail the purpose and objectives of the scientific work.
4. Refine the method of synthesis of carbon nanotubes
5. Describe the figure 4 in more detail.
6. It is desirable to add Raman spectra or IR spectroscopy of the material before and after grinding.
7. Supplement the conclusions and the abstract.
Round 2
Reviewer 1 Report
I recommend this article to be accepted.